# Are Follow-Up Blood Cultures Useful in the Antimicrobial Management of Gram Negative Bacteremia? A Reappraisal of Their Role Based on Current Knowledge

**DOI:** 10.3390/antibiotics9120895

**Published:** 2020-12-11

**Authors:** Francesco Cogliati Dezza, Ambrogio Curtolo, Lorenzo Volpicelli, Giancarlo Ceccarelli, Alessandra Oliva, Mario Venditti

**Affiliations:** Department of Public Health and Infectious Diseases, University “Sapienza” of Rome, 00185 Rome, Italy; francesco.cogliatidezza@uniroma1.it (F.C.D.); ambrogio.curt@uniroma1.it (A.C.); lorenzo.volpicelli@uniroma1.it (L.V.); giancarlo.ceccarelli@uniroma1.it (G.C.); alessandra.oliva@uniroma1.it (A.O.)

**Keywords:** follow-up blood cultures, Gram negative bacteremia, critically ill patients, antibiotic therapy

## Abstract

Bloodstream infections still constitute an outstanding cause of in-hospital morbidity and mortality, especially among critically ill patients. Follow up blood cultures (FUBCs) are widely recommended for proper management of *Staphylococcus aureus* and *Candida* spp. infections. On the other hand, their role is still a matter of controversy as far as Gram negative bacteremias are concerned. We revised, analyzed, and commented on the literature addressing this issue, to define the clinical settings in which the application of FUBCs could better reveal its value. The results of this review show that critically ill patients, endovascular and/or non-eradicable source of infection, isolation of a multi-drug resistant pathogen, end-stage renal disease, and immunodeficiencies are some factors that may predispose patients to persistent Gram negative bacteremia. An analysis of the different burdens that each of these factors have in this clinical setting allowed us to suggest which patients’ FUBCs have the potential to modify treatment choices, prompt an early source control, and finally, improve clinical outcome.

## 1. Introduction

Bloodstream infections (BSIs) represent a leading cause of death in industrialized countries, with an estimate of two million episodes and 250,000 deaths annually in North America and Europe, despite the availability of new potent antimicrobial therapies and advances in supportive care. In particular, hospital-acquired BSIs are a major cause of morbidity and mortality in intensive care units (ICU), and septic shock still represents the first cause of ICU total mortality [1]. This burden is likely to grow over the next few decades due to the increase in life-expectancy and in median number of patient comorbidity [2]. Unlike Gram positive (GP) BSIs, whose incidence rate has declined over the last few decades, Gram negative (GN) BSIs have markedly increased overtime and nowadays account for up to half of BSIs, with a mortality rate of 20–40% [3,4,5].

When considering GP-BSIs, international guidelines and consolidated evidence-based procedure bundles are available for the management of the leading pathogen species, *Staphylococcus aureus*. In this setting, follow-up blood cultures (FUBCs) are regarded as essential to document clearance of bacteremia after treatment initiation and exclude seeding [6,7,8]. On the other hand, FUBCs are mandatory in the case of *Candida* spp. BSIs in order to determine the end of candidemia and optimize treatment duration [9].

As for GN-BSIs, relevant advances in management strategies have been made in the last few years, such as the non-inferiority demonstration of 7 vs 14 day antibiotic courses [10] and of oral step-down vs continued parental therapy [11] in uncomplicated GN-BSIs. Recently, a combined approach of rapid diagnostic testing with a bundle of antimicrobial stewardship found a decrease in readmission rate and in cost per case [12]. Anyway, the management of GN-BSIs remains poorly codified and thus prone to personal clinician judgement, as compared to Gram positive settings. In a recently published scoping review, Fabre et al. proposed an algorithm for bacterial blood culture (BC) recommendations. They found that in bacteremias due to *Enterobacterales*, FUBCs are unlikely to grow unless the source of infection is endovascular or there is inadequate source control. Although with small numbers, similar results were found in *Pseudomonas* infections. The authors suggest clinical judgment to evaluate the need of FUBCs for GN. Of note, this review considered studies published from 1 January 2004 to 1 June 2019 [13], but the majority of studies addressing the topic of FUBCs in GN infections are actually subsequent to this time frame. At the time of writing, the role of FUBCs in Gram negative bacteremia (GNB) still represents an important matter of debate, with controversial results [14].

The early studies conducted have focused on the disadvantages of FUBCs, mainly represented by the risk of false positive results, prolonged hospitalization, inappropriate antibiotic use and increased cost [15,16]. Recently, the issue of FUBCs came out on top through the availability of new evidence that may have tipped the balance in favor. In this review, we aimed to examine and summarize the current knowledge on the usefulness of FUBCs in GNB, especially in the light of the reassessment of this management tool by recent studies. Moreover, we propose two guidance tools (clinical and microbiological) that summarize and graduate the recommendations for FUBCs.

## 2. Results: Review of the Literature

In 2004, Tabriz et al. published the first study where FUBCs in GNB were suggested. The authors conducted a retrospective single center study of 96 patients with at least one FUBC over 1 month (199 BC episodes without differences between GP and GN). Most FUBCs were performed within 4 days from first positive blood culture (FPBC) and during antimicrobial therapy (AT) (both 158, 79.4%). The common reasons to repeat BCs were fever, follow-up of positive BCs, and persistent leukocytosis. Positive FUBCs after FPBC were 21 (25.9%) and after a negative BC were 1.7% of cases. The conclusions were that persistent leukocytosis and fever are poor predictors of bacteremia. Then, the authors gave some general indications about when to perform FUBCs correctly, even if no definition of FUBCs was provided. They suggested FUBCs in these circumstances: 1. new septic episode, 2. suspected endocarditis, 3. follow-up of a positive BC in certain conditions that may have diagnostic and therapeutic implications, such as *S. aureus*, GNB and candidemia, 4. confirmation of response to therapy for endocarditis or other endovascular infections caused by *S. aureus*, *Enterococcus* spp., GN or other difficult-to-treat organisms because the only use of clinical data may not be reliable, 5. confirmation of diagnosis of intravascular catheter-associated bacteremia. Even if the data presented do not seem to support the above cited conclusions, this was the first time that the question of FUBCs in GNB was dealt with [17].

A summary of the main subsequent studies on the role of FUBCs in the management of GNB is reported in Table 1.

In a retrospective multicenter case-control study of 2013 [18], the authors analyzed 1068 individuals with *Klebsiella pneumoniae* bacteremia (KpB) and showed a wide prescription of FUBCs as these were performed in 80.7% of cases, while only 7.2% were found to be positive. Moreover, 53.2% of patients with non-persistent KpB underwent more than two consecutive BCs. The routine use of FUBCs was considered not justified because of the low incidence of persistent bacteremia (PB) detected. Unfavorable treatment response on the second day after the initial BCs, intra-abdominal infection, high weighted Charlson comorbidity index, and prior solid organ transplantation (SOT) were recognized as independent risk factors for persistent KpB. The authors stated that the retrospective analysis, the small sample size, and the lack of a multivariable analysis of mortality-related factors represented possible limitations of their study. Furthermore, they focused only on a specific pathogen (*K. pneumoniae*) and, although transfer to ICU was considered as an outcome, no data were available concerning the original patient allocation (ICU vs non-ICU ward) [18].

In 2016, Wiggers et al. conducted a retrospective monocentric cohort SCRIBE study on a mixed population of 1801 patients with a first episode of bacteremia caused by GP, GN or anaerobic bacteria. FUBCs were executed in 701 patients (38.9%) and PB was demonstrated in only 118 (6.6%) of the whole population. As expected, an endovascular source of infection, *S. aureus* and the inability to achieve source control in 48 h were associated with higher risk of PB. Analyzing the data provided, there were 901 GNBs (50% of the whole cohort) and 247 of them had FUBCs taken (27.4%), of which 27 (10.9%) tested positive, compared to GP bacteremias (GPBs) where BCs were repeated in 457 out of 882 patients (51.8%), with a positive yield in 90 (19.7%). Male sex, admission to a medical service, *S. aureus* bacteremia and endovascular or epidural focus were identified as risk factors for PB, but unfortunately this multivariate analysis was conducted on the whole population, rather than only on patients with GNB. Authors concluded that bacteremias caused by GNs, viridans group or beta-hemolytic streptococci are common situations in which repeat BCs offer low yield, with the related inappropriate expense. When possible, a revision of the charts regarding physician’s impression on clinical status was made and authors inferred that only 30.3% of FUBCs were drawn because of patients’ instability, a situation in which FUBCs could be suitable. Regardless of the result, 30 day mortality was significantly higher (27%) among patients undergoing repeated BCs [19].

Until 2020, the most relevant and influential article addressing the topic of FUBCs in GNBs was that conducted by Canzoneri et al. in 2017. They retrospectively analyzed 500 episodes of bacteremia, of which 383 (77%) had at least one FUBC taken. Among these 383, 206 (54%) had initial bacteremia caused by a GP organism and 140 (37%) by a GN, with an average of 2.37 FUBCs per patient. The FUBCs yielded positive in 55 (14%) of the overall population, 43 (78%) of those with GP cocci and eight (15%) with GN bacilli. The incidence of PB, defined as positive FUBC for the same original organism, was 21% in GPBs, 10% in polymicrobial bacteremias and 6% only in GNBs. Fever on the day of FUBC sampling, presence of an intravenous (IV) central catheter and end-stage renal disease (ESRD) were associated with a higher probability of PB in the whole cohort. Subgroup analysis confirmed these factors, with the adjunct of diabetes mellitus, as predictors of positive FUBC, only among subjects with GPB, while fever was the only factor associated with PB in GNB. No impact of positive FUBC on ICU admission or mortality was detectable. No clue concerning the clinical reasons for drawing FUBCs was available. The authors concluded that FUBCs may have little utility in patients with GNB, as compared to the serious negative implications of unrestrained use, represented by false positive results, longer hospital stays and increased healthcare costs [15].

In 2019, Shi et al. reported the results of a monocentric case-control study in 333 patients with bacteremic urinary tract infection (UTI): 306 (91.9%) of them had FUBCs drawn, of which 55 (18%) tested positive. Among all those that underwent FUBCs, 264 (86.3%) had a GN-related UTI with positivity in 39 (14.8%), compared to 14 (4.6%) with a GP UTI that yielded positive FUBCs in six (42.9%). Of note, four out of six of this latter group were caused by *S. aureus*. PB, defined as more than seven days of positive BCs, occurred in only six (3.3%) out of 306 patients. Several clinical and biochemical factors were associated with higher probability of PB. Eventually, four factors were selected and confirmed through multivariate analysis as independent predictors: malignancy, initial ICU admission, high c-reactive protein (CRP) level and longer time to defervescence. Among the subgroup of patients without any of these risk factors, no one had a positive FUBC. The authors concluded that, due to the low positivity rate, liberally prescribed FUBCs have little utility in the management of bacteremic UTI [20]. However, the results of this study might be limited by the design focused on only one clinical syndrome (UTI), the mixed causative agents considered (including a not negligible proportion of *S. aureus*) and the lack of data regarding the effect of pathogen antibiotic susceptibilities on FUBC results, clinical reasons for drawing FUBCs and original admission service of the patients.

In the same year, a study conducted in a pediatric hospital of Tokyo firstly questioned the conception of the usefulness of FUBC in the setting of GNB. Uehara et al. enrolled 99 children with GNB, with a median age of two years. The most frequent underlying diseases were SOT (21.2%), malignant neoplasm (17.2%) and kidney/urinary tract malformation (15.2%); a central venous catheter (CVC) was in place in 57% of patients. Twenty-one patients (21.2%) had positive FUBCs, with *Klebsiella* spp. and *Escherichia coli* being the most represented pathogens. Interestingly, no cases of positive FUBCs emerged among patients with UTI. Multivariate analysis revealed the presence of CVC and resistance to empirical therapy as significantly associated with PB. More importantly, the authors reviewed clinical charts and reported that the positive yield of FUBCs promoted a treatment modification in 57% of patients, which included optimization of antibiotic therapy and/or removal of medical devices [21].

In 2020, the prior view of FUBCs as a tool of little utility in patients with GNB, counterpoised to many clinical and economical drawbacks, underwent a systematic reassessment. Completely different results from those obtained by Canzoneri et al. [15] were in fact reported by Giannella et al. in a single center, retrospective cohort analysis of 1576 patients with GNB [22]. As in previous studies, FUBCs were prescribed based on personal clinical judgement rather than systematically. Nevertheless, FUBCs were performed in only 278 (17.6%) patients but demonstrated a high rate of PB: 107 (38.5%). Patients that underwent FUBCs were younger, with a lower Charlson comorbidity index, but more frequently immunocompromised, admitted to ICU, with a hospital-acquired GNB, and with a non-urinary source of infection, compared to those without FUBCs performed. Furthermore, patients with FUBC taken had higher initial severity of GNB clinical pictures according to SOFA (Sequential Organ Failure Assessment) score and septic shock criteria, higher frequency of carbapenem-resistant enterobacteriaceae (CRE) isolation and of inappropriate empirical therapy. Thus, as a matter of fact, the patient complexity seemed to progressively rise from those without FUBCs drawn, those with FUBCs, to those with positive FUBCs. That is to say that, for the first time since the topic of FUBCs in the setting of GNB has been debated, the authors provided elements to interpret the mechanism by which physicians currently use FUBCs. Interestingly, taking into account the higher complexity of patients that underwent FUBCs, the execution of FUBCs was followed by increased rate of source control, infectious disease consultation and longer treatment duration. Thus, performance of FUBCs appeared to act for physicians as an incitement to more careful management. At the same time, Giannella et al. demonstrated that FUBCs had a favorable impact on patient outcome, an effect probably linked to prompt source control. In fact, through multivariate analysis, FUBCs, along with UTI origin of BSI, source control and active empiric therapy, resulted as independent factors protective from all-cause 30 day mortality. The authors concluded that future prospective studies with a systematic use of FUBCs in GNB are necessary in order to better identify the settings where FUBCs could be cost-effective [22].

Similar favorable results of FUBC use were obtained by Maskarinec et al. in an observational study of 1702 prospectively enrolled inpatients with monomicrobial GNB [23]. FUBCs were drawn in 1164 patients (68%) and more commonly in patients with *Pseudomonas aeruginosa* and *Serratia* spp. (80%). PB was detected in 228 (20%). Patients with PB had a lower probability of having under effective antibiotic treatment (with higher rates of fluoroquinolone and/or carbapenem-resistant isolates in FUBCs) and higher probability of being a transplant recipient, hemodialysis dependent, having a cardiac device, recent corticosteroid use, a malignancy, or an endovascular source of infection. Bacteremias caused by *Serratia* (32%, 95% CI 24–44%) and *Stenotrophomonas maltophilia* (52%, 95% CI 32–72%) had the higher rate of persistence. Patients’ clinical outcomes were also evaluated. Relevantly, the regression model showed that obtaining FUBCs was associated with decreased rates of both all-cause and attributable mortality. This result was also confirmed in a species-specific analysis performed for *E. coli* and *K. pneumoniae* and in a sensitivity analysis that excluded all deaths occurring in the first 48 h. On the contrary, PB implied a nearly double all-cause and attributable mortality relative to those with negative FUBCs, and similar to those without FUBCs drawn. The probability of PB was estimated through a risk scoring system and finally, an endovascular source of infection was identified as the only breakpoint separating high and low rates of FUBC positivity. The authors concluded that FUBCs have clinical utility in detecting patients with increased risk of poor outcome, and that could benefit from additional diagnostic and therapeutic interventions. Considering the low rate (2%) of false positivity FUBCs and the little difference in duration of antibiotic treatment (2 days), this study also scaled back the traditional concerns of increased healthcare costs and antimicrobial prescription usually attributed to FUBCs [23].

On the other hand, Jung et al. from South Korea recently presented a retrospective observational cohort study conducted on 1481 cases of GNB. FUBCs were widely performed (86.2%), while positivity resulted in 122 (9.6%) [24]. Comparing the clinical characteristics of those that underwent FUBCs and those that did not, female gender, neutropenia, hematologic malignancy, presence of an intravascular device and of an extended spectrum β-lactamase (ESBL)-producing organism were more represented in the first group, while a biliary source was more common in the latter. No difference was detected in terms of incident mortality between FUBCs drawn and not. The comparison between patients with positive and negative FUBC yield was made by sub-stratification according to eradicable and non-eradicable source of infection. Several factors were identified and included in a predictive scoring model if independently associated with FUBC positivity through a multivariate logistic regression. Results indicated that, in the case of a removable source of infection, if there is appropriate management (early source control and appropriate therapy) followed by a favorable clinical response (quick SOFA score <2), performing FUBCs adds little value. Furthermore, even in a non-eradicable setting, the administration of effective treatment corresponded to 95% probability of negative conversion, regardless of the underlying disease, offending pathogen, or treatment response. Author conclusions were that FUBCs can be avoided in most uncomplicated cases of GNB and could be considered selectively in high-risk patients. In addition to the retrospective nature of the study, it should be underlined that neither the patients’ outcome nor the relative impact of FUBCs’ execution/results were evaluated. In addition, mortality was not taken into consideration in the comparison between positive and negative results of FUBCs and no data concerning ICU vs non-ICU ward allocation were provided [24].

Mitaka et al. conducted a retrospective multicenter observational study in all adults with at least one BC positive for GNs admitted between January 2017 and December 2018 [25]. A total of 463 patients were included; of these, 306 (66%) had FUBCs performed at least once. The results showed positive FUBCs in only 10% of patients. The authors found a correlation between positive FUBCs and the following risk factors: ESRD, presence of intravascular devices, and bacteremia due to ESBL-producing organism or CRE. The yield of positive FUBCs in patients without the risk factors was 1.6%, compared to 14.8% in the presence of ≥1 risk factor. The authors concluded that FUBCs may not be necessary for all GNBs, but only in the presence of risk factors [25]. In addition to the retrospective design and the lack of standardization of decision making regarding FUBCs, other limitations of the study included that authors only analyzed the positivity or negativity of the first FUBC without checking the possibility of intermittent bacteremia and no information about clinical outcomes and therapeutic change based on the results of FUBCs was provided.

In 2020, Spaziante et al. conducted a retrospective single center observational study on 307 patients admitted to a multidisciplinary ICU in 2017 [26]. Sixty-nine patients (22.4%) presenting with at least one GNB episode for a total of 107 episodes were included in the study. Exclusion criteria were the occurrence of fungemia, GP or mixed GP/GN bacteremic episodes. FUBCs were defined as BC performed within 48 h from the beginning of antimicrobial therapy (AT) and then every 24–72 h after FPBCs. PB was defined as repeatedly positive BCs for GNs after ≥96 h of appropriate AT and ≥48 h after removal of all potentially infected endovascular indwelling devices. Twenty-nine GNB episodes (27.1%) were excluded from the study because no FUBCs were performed. Eventually, 28 (35.9%) out of 78 GNB episodes were diagnosed as PB. Under these circumstances, septic thrombosis (ST) was the hematogenous source of infection in approximately half of the cases, resulting in a significant association with positive FUBCs (*p* < 0.001). On the other hand, negative FUBCs were associated with primary bacteremia (*p* < 0.001). As part of the retrospective design, this is the only paper entirely focused on critically ill patients. In particular, the study was conducted in an ICU that is a reference center for polytrauma and, based on the aforementioned results, the authors hypothesized that frequent deep venous thrombosis occurring near to bone fractures may provide a suitable medium for microbial seeding for GNB originating from other body sites [26].

## 3. Discussion

GNB still represents an extremely relevant cause of morbidity and mortality in hospitalized patients, especially in ICU settings. Therefore, it is crucial to achieve an optimization of patient care, which, at the same time, could reduce mortality rates and meet the growing demands of antimicrobial stewardship and cost control. In this regard, the use of FUBCs in patients with GNB has represented a contentious topic in the past few years and especially in the very last period.

Indeed, unnecessary FUBCs may cause patient discomfort and carry the risk of false positive results. According to previous reports, as many as 90% of all BCs grow no organisms and of the approximate 10% that do grow organisms, almost half are considered contaminants (false positives) [27]. Thus, given a constant rate of contamination, performing more FUBCs may result in a higher chance of encountering contaminant organisms, and consequently, in increased costs and patients discomfort, longer hospital stays, unnecessary consultations, and inappropriate antimicrobial therapy [15,16]. Of note, this reasoning includes an inherent fallacy. Although theoretically acceptable in cases of GP yield, the growth of GN bacteria in BCs should always be regarded as relevant and never, or just anecdotally, be considered as a contamination [28].

On the other side, performing FUBCs may have a relevant impact on patient management and outcome, reducing mortality rates. When FUBCs are performed in more severe patients with comorbidities and without adequate infection source control, in cases of bacteremia due to multidrug resistant (MDR) GNs and without an appropriate empiric therapy, the positive or negative results may guide the clinician to the correct decision about type and duration of antibiotic therapy [22,23].

Not surprisingly, the evidence concerning the usefulness of FUBCs underwent a progressive shift in the last few years, going from a restrictive to a selective approach. Taking a look at Table 1, it appears that papers with a higher rate of FUBCs performed, more frequently found no evidence of benefit in contrast to those that applied FUBCs more selectively. In fact, Kang et al. [18] and Jung et al. [24] performed FUBCs in GNB in 81% and 86% of the cases, respectively, as compared to Maskarinec and Giannella that used this tool only in 68% and 17% of cases, respectively. Likely, Spaziante et al. [26] found a high rate of PB by only performing FUBCs in high-risk ICU patients with GNB. Furthermore, the inclusion in the analysis of a mixed GP and GN population was also associated with pessimistic results. Therefore, it seems that the more refined the selection of patients in which to draw FUBCs, the more evident the benefits they bring. Indeed, the results of our literature review show that, while in some subgroups of patients the use of FUBCs may not translate into a clear benefit, it is possible to identify several situations where the application of this tool may steer the clinical decision making in the correct way. Thus, selection is a sticking point in this topic.

For the purposes of performing rational FUBCs, physicians well trained in infectious diseases should be available in all settings where risk factors for positive FUBCs are present. For this reason, ICU patients might be evaluated from dedicated infectious disease consultants in order to avoid unnecessary BCs [22,26]. The study written by Ceccarelli et al. shows an example of this management [29]. In fact, they reported some cases of GN-related septic thrombosis (ST) with indolent clinical course and long-term positive BCs despite adequate antibiotic treatment as one possible exception to the restricted use of FUBCs. In these cases, FUBCs allowed the determination of the correct treatment duration, representing a tool of critical importance for patient management [29]. The same group of authors further stressed this concept in a case series of 13 critical care patients with ST caused by GN bacilli (Figure 1): this disease was characterized by PB despite prompt source control and appropriate antibiotic treatment, an indolent clinical course and, even more important, a rapid defervescence with normalization of procalcitonin (PCT) values preceding bacteremia clearance. This phenomenon was interpreted as a mechanism of immune tolerance [30].

Based on the fascinating hypothesis that FUBCs in BSIs due to GNs could be part of clinical practice, we tried to investigate which conditions make FUBCs either necessary or unfounded. Figure 2 and Figure 3 show clinical and microbiological risk factors for PB, respectively. Rows and columns of each figure intersect in a colored square and every color means a risk threshold of PB, from green (FUBCs highly recommended), light green (moderate recommendation), yellow (weak recommendation) to red (avoid FUBCs), of various combinations of all risk factors recognized in this review.

In order to point out the right setting where FUBCs should be prescribed (otherwise when they are not warranted), we created Figure 2 and Figure 3 analyzing the risk factors for persistent bacteremia found in the revised articles. On the other hand, we identified settings where FUBCs are moderate or even weakly recommended based on risk factors cited by less authors/articles and our judgment. For these reasons, a clinical decision on a case-by-case basis is needed to judge when to perform FUBCs.

In general, FUBCs should be always considered in critically ill patients because they frequently present multiple risk factors that may account for resistant or persistent GNB: intravascular catheter, antibiotic resistant pathogen or an occult source of infection that requires control. As an example, FUBCs are warranted in patients with an endovascular source of GNB that represents the single most important indication to this procedure, even in some instances where a biomarker of active infection, such as PCT, is decreased to negative values. In fact, Spaziante et al. stressed this point by analyzing Gram negative ST where positive FUBCs played a crucial role in patient outcomes (Figure 1). Along the same line, the presence of a non-eradicable infection source appears to be a condition in which FUBCs should be performed, especially in patients that require ICU, with persistent fever or as a moderate recommendation (light green) in individuals afflicted with ESRD on hemodialysis [23,24,25]. FUBCs might also be indicated in persistently febrile patients with primary bacteremia, long term intravascular catheter or urinary tract infection, to a lower degree of evidence. Additionally, we suggest that even in some instances of a lower positivity rate of FUBCs, a clinical decision might be reached in cases of light green and yellow squares both in Figure 2 and Figure 3. For this reason, in the case of patients without a clear clinical indication to perform FUBCs as shown in Figure 2, they should be checked for microbiological risk factors, as shown in Figure 3. To this end, in our opinion, FUBCs might be performed even in instances of microbiological risk factors alone. In fact, taking clinically stable patients with UTI as an example (yellow to red squares in Figure 2), instances where the column of MDR microorganisms’ etiology and the row of ineffective therapy intersect in a green square might represent a recommendation to perform FUBCs [20,21,24,25].

Getting to the point, if Figure 2 recommends FUBCs, it deals with a “green light” and they should be performed in any case. Additionally, when FUBCs are not indicated by clinical risk factors, clinicians should check the presence of microbiological risk factors, as shown in Figure 3. In the case of neither clinical nor microbiological risk factors, we are in front of a “red light” and FUBCs are not warranted.

A possible limitation of Figure 2 and Figure 3 is that the definitions of risk factors are often different between studies or even not provided at all. For instance, the dosage and duration of corticosteroid treatment with a significant immunosuppressive effect are not clearly defined in the literature as they are considered to depend on the characteristics of the patient and underlying disease [31].

Finally, progressive acquisition of resistance during antimicrobial therapy through the selection of a hidden resistant subpopulation (named hetero-resistance) is a growing concern in the case of persistent bacteremia [32]. Future studies should elucidate the possible role of FUBCs in early detection and management of this increasingly appreciated mechanism of resistance.

**Figure 2 antibiotics-09-00895-f002:**
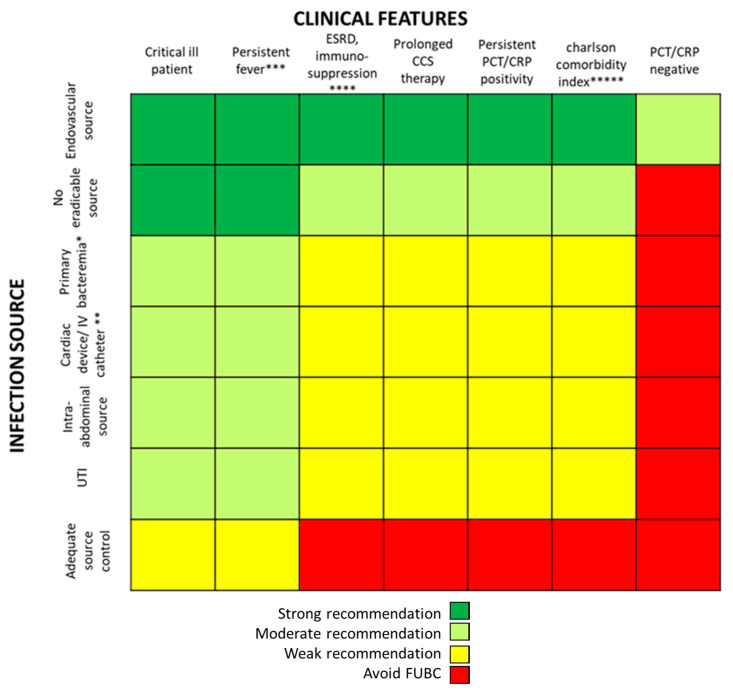
Analysis of recommendations for FUBC sampling in the setting of Gram negative bacteremia on the basis of clinical features and source of infection. **Note:** infection source: * when apparently there is not an infection source; ** if another infection source is presented; *** fever for more than 72–96 h; **** malignancy, solid organ transplantation; ***** Charlson comorbidity index ≥ 6. References: critical ill patient [20,22,23,25,26]; persistent fever [15,20,26]; ESRD, immuno-suppression [15,18,20,23,24,25]; prolonged corticosteroid (CCS) therapy [23]; persistent PCT/C-reactive protein (CRP) positivity [20]; Charlson comorbidity index [18]; PCT/CRP negative [20]; endovascular source [19,23,24,26]; no eradicable source [22,24,25]; primary bacteremia [26]; cardiac device/intravenous (IV) catheter [15,21,23,24,25]; intra-abdominal source [18]; UTI [20,23,25]; adequate source control [24]. **Abbreviations:** UTI, urinary tract infections; ICU, intensive care unit; ESRD, end stage renal disease; CCS, corticosteroids; PCT, procalcitonin; CRP, C-reactive protein.

**Figure 3 antibiotics-09-00895-f003:**
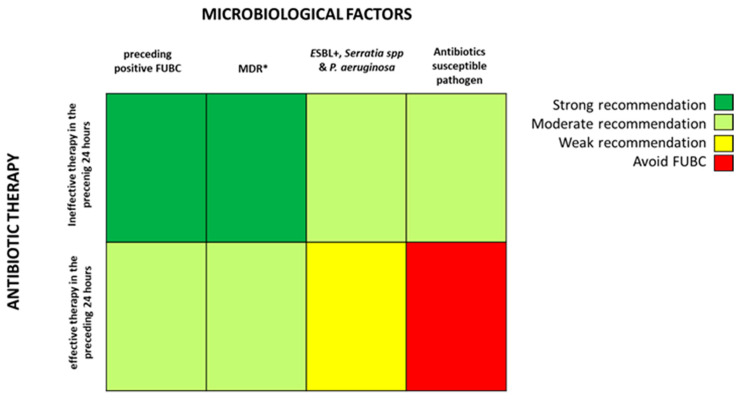
Analysis of recommendations for FUBC sampling in the setting of Gram negative bacteremia on the basis of microbiological factors and efficacy of antibiotic therapy. **Note:** * MDR definition by Magiorakos AP et al., 2012 [33]; ESBL, extended-spectrum beta-lactamases. References: ineffective therapy in the preceding 24 h [18,21,22,23]; effective therapy in the preceding 24 h [18,21,22,23,24]; preceding positive FUBC [26]; MDR [22,23,25,26]; ESBL+, Serratia spp. and P. aeruginosa [23,24,25]; antibiotic susceptible pathogen [22,23,25,26].

## 4. Materials and Methods

We searched in the Pubmed database for articles addressing the use of FUBCs in patients with GNB. The following search strategy was adopted: “((FUBCs) or (Follow-up blood cultures) or (follow-up blood culture) or (repeat blood cultures)) and ((gram-negative) or (gram negative rod)) and ((bacteremia) or (BSI) or (bloodstream infection))”. The research yielded 102 results. Two reviewers independently assessed the titles and abstracts to identify papers that fulfilled the inclusion criteria: (1) clinical studies; (2) studies that included human subjects; and (3) studies that evaluated the utility of FUBCs in patients with GNB. Full texts of studies assessed as relevant or unclear were evaluated. Studies that only discussed either GNB or FUBCs were excluded. We also examined the bibliographic references of articles to identify any relevant studies that were not identified in the initial literature search. Eleven articles were selected, compared, and critically evaluated (Figure 4).

## 5. Conclusions

The usefulness and the drawbacks of FUBCs in GNBs have been largely investigated in the last few years. Of course, some clinical and microbiological factors define the settings where FUBCs exert their maximum capacity to detect PB. Expert clinicians and a correct selection of high-risk patients could make the difference in terms of the efficiency of this diagnostic tool. Furthermore, a targeted and optimized selection of the occasions where to draw FUBCs may also provide a positive impact on patients’ management and outcomes. Of note, valuable insights about outcomes of patients where FUBCs were performed remain poor and should be further investigated.

## Figures and Tables

**Figure 1 antibiotics-09-00895-f001:**
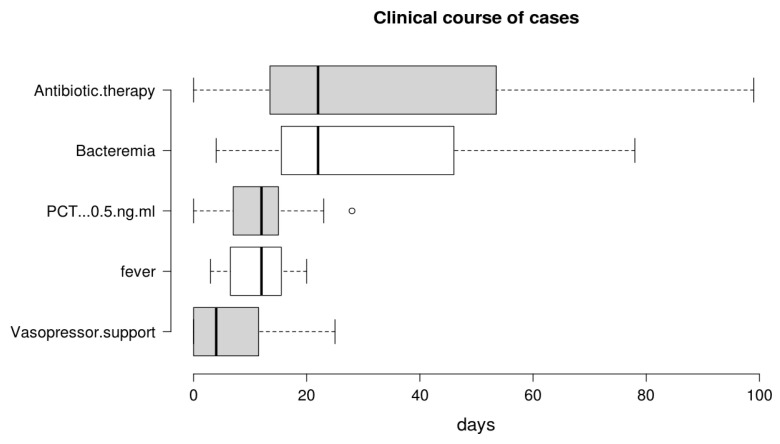
Duration of bacteremia and clinical course features of Gram negative septic thrombosis in critically ill patients, modified from Spaziante et al. [30]. This figure shows that bacteremia may persist despite clinical improvement (fever disappearance, negative procalcitonin (PCT) values and no vasopressor support); under these circumstances, FUBCs may remain the only driver of antibiotic therapy.

**Figure 4 antibiotics-09-00895-f004:**
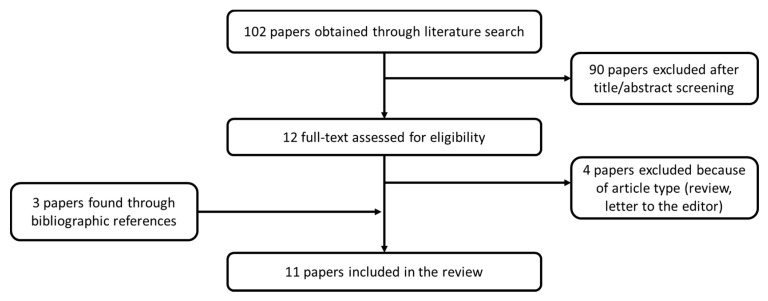
Flow chart of article selection process. References of eleven selected articles: [15,17,18,19,20,21,22,23,24,25,26].

**Table 1 antibiotics-09-00895-t001:** Summary of the main studies on follow up blood cultures (FUBCs) in Gram negative bacteremia.

Study: First Author, Year Design	✔ Inclusion Criteria ✕ Exclusion Criteria	FUBC Definition N of FUBCs/N of Patients (%)	Positive FUBC /N of FUBCs (%)	Positive ICU FUBCs/ICU Patients (%)	Results and Conclusions	Limitations
Kang, 2013 retrospective case-control [18]	✔ Age ≥ 18 ✔ First episode of KpB ✕ Polymicrobial ✕ Recurrent KpB	BC drawn after more than 2 days from the index BC 862/1068 (81)	62/862 (7.2)	NA	Risk factors for persistent KpB included in a clinical score: - unfavorable treatment response - intra-abdominal infection - high CCI - SOT Routine FUBCs not justified.	Retrospective Only one pathogen considered No ICU data No multivariate for mortality risk factors
Wiggers, 2016 retrospective cohort [19]	✔ Age ≥ 18 ✔ First episode of GNB ✕ Recurrent GNB	BC drawn 2–7 days from the index BC 247/901 (27.4) [GN only considered]	27/247 (10.9) [GNB only considered]	NA	Increased 30 days mortality in patients undergoing FUBCs, regardless of the result Repeated BC in GN bacteremia offer low yield	Retrospective Mixed population of GP and GN No ICU data Small sample with PB (defined as positive BC 2–7 days after the index BC
Canzoneri, 2017 retrospective case-control [15]	✔ Age ≥ 18 ✔ One positive BC ✕ Fungemia ✕ Potential contaminants	BC drawn after at least 24 h from the index BC 383/500 (77) [GP + GN considered]	8/140 (6) [GNB only considered]	18/165 (10.9) [GP + GN considered]	PB more common for GPs (21%), than polymicrobial (10%), than GNs (6%) FUBCs have little utility in patients with GNB	Retrospective Mixed population of GP and GN Polymicrobial infections included Small number of patients with PB
Shi, 2019 retrospective case-control [20]	✔ Age ≥ 18 ✔ Bacteremic UTI ✔ At least one FUBC ✕ Non-urinary source of bacteremia	More than one separate BC taken more than 24 h after the index BC 306/333 (92) [GP + GN considered]	39/264 (14.8) [GN only considered]	20/55 (36.4)	Not recommended routine FUBC Predictors for positive FUBC in bacteremic UTI: malignancy, initial ICU admission, high CRP level and longer time to defervescence	Retrospective Mixed population with GP and GN Only UTIs considered
Uehara, 2019 retrospective observational [21]	✔ Age ≤ 18 ✔ GN B ✕ Polymicrobial ✕ FUBCs ≤ 24 h from the index BC ✕ AT started ≥ 48 h from the index BC	BCs taken over 24 h from the index BC 99/137 (72.3)	21/99 (21.2)	NA	FUBC may still be useful in the management of GNB in children Presence of a CVC and resistance to empirical antibiotics were risks for positive FUBCs	Retrospective Small sample Possible selection bias
Giannella, 2020 retrospective cohort [22]	✔ Age ≥ 18 ✔ GNB ✕ Polymicrobial ✕ Potential contaminants ✕ Death ≤ 72 h after index BC ✕ Unavailable clinical data	BCs drawn between 24 h and 7 days after the index BC 278/1576 (17.6)	107/278 (38.5)	21/126 (16.6)	FUBCs drawn in more severe, high risk, antibiotic resistant and initially inappropriately treated patients In this context, FUBCs execution associated to higher rate of source control, ID consultation and lower 30-day mortality	Retrospective Single center
Maskarinec, 2020 prospectively enrolled cohort [23]	✔ Age ≥ 18 ✔ GNB ✕ Polymicrobial ✕ Death ≤ 24 h after index BC	BCs drawn from 24 h to 7 days from index BC 1164/1702 (68.4)	228/1164 (19.6)	4/41 (9.8)	FUBCs drawn in high risk patients Obtaining FUBCs associated with decreased all-cause and attributable mortality Positive FUBCs associated with increased all-cause and attributable mortality	Poor data on patients’ clinical status at FUBCs collection No data on management changes based on FUBC results
Jung, 2020 retrospective observational cohort [24]	24GNB ✕ Age﹤18 ✕ Death ≤ 48 h after index BC ✕ Polymicrobial ✕ Different species from the index BC identified by FUBC	BCs drawn 2–7 days from index BC 1276/1481 (86.2)	122/1276 (9.6)	NA	FUBCs can be avoided in most uncomplicated cases of GNB and could be considered selectively in high risk patients Two clinical scores for patients with eradicable and non-eradicable source of infection	Retrospective Not evaluated the impact of FUBCs on patient outcome
Mitaka, 2020 retrospective multicenter observational [25]	✔ Age ≥ 18 ✔ GNB Potential contaminants	BC draws after at least 24h of AT 306/463 (66.1)	28/306 (9.2)	18/130 (13.9)	RF for positive FUBCs: ESRD on hemodialysis, intravascular device, ESBL or carbapenemase-producing organism Higher yield of positive FUBCs in patients with ≥ 1 RF Routine FUBCs are not necessary	Retrospective Only the first FUBC analyzed 26% FUBCs not performed without standardizing the decision
Spaziante, 2020 retrospective observational study [26]	✔ Age ≥ 18 ✔ GNB ✔ FUBC ≤ 24–72 h from the FUBC or ≤ 48 h from the beginning of AAT ✕ Polymicrobial	BCs done within 48 h from the beginning of AT and then every 24–72 h after FUBCs 78/107 (73) [GNB episodes from 69 patients]	28/78 (35.9) [patients]	[All patients were in ICU]	Septic thrombus infection was the source in 14 (50%) cases of GNB-PB. A MDR isolate was in 60 BCs (76.9%) FUBCs represent a useful tool in the management of GN-PB, especially if caused by STI	Retrospective Only ICU patients from polytrauma ICU Small sample

**Abbreviations**: AT, antimicrobial therapy; BC, blood culture; BSI, blood stream infection; CCI, Charlson comorbidity index; CRP, C-reactive protein; CVC, central venous catheter; ESBL, extended spectrum β-lactamase; ESRD, end stage renal disease; FUBC, follow-up blood cultures; GN, Gram negative; GNB, Gram negative bacteremia; GP, Gram positive; ICU, intensive care unit; KpB, *Klebsiella pneumoniae* bacteremia; MDR, multidrug resistant; NA, not available/not applicable; PB, persistent bacteremia; RF, risk factor; SOT, solid organ transplant; UTI, urinary tract infection.

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
