# Peer review of "Are Follow-Up Blood Cultures Useful in the Antimicrobial Management of Gram Negative Bacteremia? A Reappraisal of Their Role Based on Current Knowledge"

_antibiotics, 2020, doi:10.3390/antibiotics9120895_

Round 1

Reviewer 1 Report

Introduction:  The authors state set up this topic as a controversy, however they do not really describe the downside to obtaining FUBC, thus hard to understand the controversy just from the introduction.  Would consider bringing in some of the cons to obtaining FUBC in GNB in the introduction to make the reader more interested from the beginning. 

Line 66 - the time period of published studies reviewed seems to be through 2020.

Line 71 states "leaned towards a negative opinion", this needs to be clarified as to what they mean by this. I believe they are saying earlier studies suggest less are needed but later studies suggest more are needed, but this is not necessarily what they end up concluding. 

Table 1: Gianella row - in 2nd to last box, I am not sure what they are saying here. Then the last bullet in this box states FUBCs improved patient management and outcomes; would be more specific and say there was an association of obtaining FUBC with what specific outcomes.  The limitations box says "no genotypic analysis in persistent BSI", but i am not sure this is a limitation that is relevant here.

Mitaka row - in the last box, the last bullet states no evaluation of outcome : FUBC vs no FUBCs, but also not sure if this is a relevant limitation as the outcome of interest was positive FUBC, therefore it would not be compared to no FUBC.  Or they may be talking about another outcome not specified in the table?

Line 111 - "systematic approach" is probably not the best way to describe a retrospective multicenter case-control.  Would consider a different leading sentence for this paragraph or just dropping this sentence.

Line 216-217 - Does the article cited actually make this conclusion? Because it would seem that the higher complexity of the patient would be the inciting cause for more careful management, which would include FUBCs, and not the obtainment of the FUBC that lead to the more careful management. 

Line 352 - was not sure what ST meant until i went back to the paper cited in this paragraph. 

Figures 2 and 3 could very helpful for clinical guidance and i would recommend talking about these two tools being developed as the point of your review.  These are very nice and would even be interested in seeing these guidance tools validated in a prospective population, in the future.

There are several uses of the word "anyway" as a transitional word at the beginning of a sentence that just does not seem to fit, most of the time there is no need for a transition there.

Line 442-443, i am not really sure whether this sentence is true with respect that this was "largely debated" since it seems that there has been a trend of suggest less FUBC in GNB towards a more tailored approach, which is a consistent natural trend over time for any intervention.  I am also not sure what they mean by "last period".

Reviewer 2 Report

The authors reviewed a few recent articles regarding the use of follow-up blood cultures (FUBC) for Gram-negative bacteremias. It remained controversial to apply FUBC for Gram-negative species from these studies; therefore, a review article would provide value in this field. The summary for the reference articles is clear. However, the analysis should be improved for clarity. The authors analyzed different conditions when FUBC is appropriate and summarized them in Figure 2 and Figure 3. I think these tables are of great value for clinicians; however, it would be beneficial to provide more clear definitions and references in these tables. It is unclear if these definitions or classifications are based on literatures or authors’ judgment.

  • The definitions of moderate recommendation weak recommendation are not clear.
  • Are the colored squares supported by the cited references or inferred by the authors? I would recommend that the authors to place the reference numbers in each colored square to show where the arguments and suggestions come from.
  • Provide the definitions of clinical features and infection sources more clearly. For example, how long is the prolonged corticosteroids therapy? This should be checked for each entry. The authors should either provide the definitions or cite references.

In figure 4, the refences for the 11 selected papers should be cited. From table 1, I only found ten references.

Reviewer 3 Report

"Are follow-up blood cultures useful in the antimicrobial management of gram-negative bacteremia? A reappraisal of their role based on current knowledge" by Venditti Mario et all is addressing important questions. Overall, this is an interesting work/review, although not 100% clear conclusion/s. I would stress more “antibiotic effect” in the whole discussion, considering Journal where authors intend to publish.  To give an example, I would suggest the following changes and reference/s:

Figure 3: I would reconsider the “Weak recommendation” (yellow color) in the upper row, 4th column and replace with the moderate recommendation in the context of  FUBC discussion-> Authors should comment possibility of acquiring (progressive) resistance during therapy of persistent bacteremia, unfortunately also very common, (sometimes detected/named as emerging “hetero-resistance”) …This would favor the introduction of FUBC, especially in the cases where the source of infection is a reservoir of microbiota-rich medium (intestine, urinary tract…). For heteroresistance, please use the reference https://www.frontiersin.org/articles/10.3389/fmicb.2020.01820/full

Lane 78 Literature not “ilteratire

79 reference

84 were (:) missing:

Round 2

Reviewer 2 Report

The authors have addressed all my questions and provided clear definition and references for the tables. Check line 436 "PCT/CRP negative []", does it have no references or the authors forgot to insert it.